# Patterns of Antibiotic Prescription in Colombia: Are There Differences between Capital Cities and Municipalities?

**DOI:** 10.3390/antibiotics9070389

**Published:** 2020-07-08

**Authors:** Jorge Enrique Machado-Alba, Luis Fernando Valladales-Restrepo, Andrés Gaviria-Mendoza, Manuel Enrique Machado-Duque, Albert Figueras

**Affiliations:** 1Grupo de Investigación en Farmacoepidemiología y Farmacovigilancia, Universidad Tecnológica de Pereira-Audifarma S.A, 660003 Pereira, Colombia; lfvalladales@utp.edu.co (L.F.V.-R.); angaviria@utp.edu.co (A.G.-M.); memachado@utp.edu.co (M.E.M.-D.); 2Grupo de Investigación Biomedicina, Facultad de Medicina, Fundación Universitaria Autónoma de las Américas, 660003 Pereira, Colombia; 3Departament de Farmacologia, Terapèutica i Toxicología, Universitat Autònoma de Barcelona, 08193 Barcelona, Spain; albert.figueras@gmail.com

**Keywords:** antibiotic prescription, AWaRe classification, Colombia, rural/urban consumption, fluoroquinolones, cephalosporins, drug utilization study

## Abstract

The use of antibiotics is the most important modifiable risk factor for the development of microorganism resistance. A cross-sectional study of outpatients receiving antibiotic prescriptions registered in a population database in Colombia was conducted. The characteristics of the consumption in capital cities and small municipalities was studied and the AWaRe classification was used. AWaRe classifies antibiotics into three stewardship groups: Access, Watch and Reserve, to emphasize the importance of their optimal use and potential harms of antimicrobial resistance. A total of 182,397 patients were prescribed an antibiotic; the most common were penicillins (38.6%), cephalosporins (30.2%) and fluoroquinolones (10.9%). ‘Access’ antibiotics (86.4%) were the most frequently prescribed, followed by ‘Watch’ antibiotics (17.0%). Being 18 or older, being male, living in a municipality, having one or more comorbidities and urinary, respiratory or gastrointestinal disorders increased the probability of receiving ‘Watch’ or ‘Reserve’ antibiotics. Penicillin and urinary antiseptic prescriptions predominated in cities, while cephalosporin and fluoroquinolone prescriptions predominated in municipalities. This analysis showed that the goal set by the WHO Access of mainly using Access antibiotics is being met, although the high use of Watch antibiotics in municipalities should be carefully studied to determine if it is necessary to design specific campaigns to improve antibiotics use.

## 1. Introduction

The discovery of antibiotics remains one of the most important recent scientific advances in human health as it has increased the life expectancy of the population. However, the proportion of infections caused by resistant bacteria and by those with new patterns of resistance has been increasing for some years [1,2]. It is estimated that more than 70% of pathogenic bacteria are resistant to at least one antibiotic [3]. The use of antibiotics, especially broad-spectrum antibiotics, is the most important modifiable risk factor for the spread of resistance [1,4,5]. Other related factors include the excessive use of antibiotics in agriculture, their misuse in veterinary medicine, their inadequate prescription in human medicine, growing globalization, and drug counterfeiting [3,6].

Infections caused by resistant microorganisms are associated with an increased risk of morbidity, complications, mortality, health services use, and increased costs [3,7,8]. Infectious diseases are currently the second leading cause of death worldwide, the third in developed countries and the fourth in the United States [9]. It is estimated that worldwide, 17 million people die each year from bacterial infections [9]. In the United States, approximately 2 million people contract antibiotic-resistant bacterial infections each year, and 23,000 of them die as a result of these infections [9]. In the United States and in European countries, 67.9% of the total disability-adjusted life year (DALY) per 100,000 were due to infections caused by four antibiotic-resistant bacteria: *E. coli* resistant to third-generation cephalosporins, Methicillin-resistant *S. aureus*, carbapenem-resistant *P. aeruginosa*, and third-generation cephalosporin-resistant *K. pneumoniae* [10].

Proper prescription of antibiotics is essential to reduce resistance [1] and ensure the long-term availability of effective treatments for bacterial infections [7], and it is currently a public health priority [11,12]. However, despite the growth of antimicrobial resistance, the approval of new antibiotics has decreased by 90% in recent years in the United States due to the high cost of developing these drugs and the rapid evolution of resistance [3]. It is evident that there is currently an urgent need to develop new antibiotics to avoid returning to the pre-antibiotic era [9].

However, it is unlikely that this problem can be solved solely by the development of new antimicrobial drugs because the threat of resistance will always accompany any new drugs introduced for clinical use. Therefore, it is essential to implement rational drug use programs, improve targeted antibiotic therapy, and establish preventive measures and faster diagnostic tools, among other strategies, as these are the only ways to preserve antibiotics for future generations and ensure a healthy future for the world population [2].

The World Health Organization (WHO) developed an instrument that seeks to improve the quality of antibiotic prescriptions to decrease the spread of resistant microorganisms and reduce adverse reactions and costs. The AWaRe tool classifies antibiotics into three groups: Access (for example: penicillins, beta lactams with beta-lactamase inhibitors, first-generation cephalosporins, tetracyclines, aminoglycosides, lincosamides, among others), Watch (for example: second, third or fourth generation cephalosporins, macrolides, carbapenems, fluoroquinolones, among others) and Reserve (for example: monobactams, fifth generation cephalosporins, polymyxins, glycopeptides, among others) [13,14,15]. Antibiotics in the Access group are those that should initially be used for the most common and severe infections, are narrow spectrum, are and less expensive; antibiotics under surveillance (Watch) should be used in moderation due to the relatively high risk of resistant strains; and Reserve antibiotics are to be used for the treatment of infections by microorganisms resistant to multiple antibiotics [13,14,15]. However, some limitations of the AWaRe tool must be recognized, including that not all classes of antibiotics are categorized, requires local adaptation and updates over time [13]. The objective of the WHO is to increase the proportion of global consumption of antibiotics in the Access group by at least 60% and reduce the use of antibiotics in the Watch and Reserve groups, which are associated with an increased risk of resistance [13,14]. Therefore, the tool could help to estimate the use of broad and narrow spectrum antibiotics, which facilitates the monitoring and optimization of their use [13].

This becomes even more important, when it has been established that the general consumption of antibiotics increased 39% between the years 2000 to 2015, where most of the dispensations occurred in primary care centers [16]. Differences in antibiotic prescription habits between countries have been established [17,18], also between different geographical regions of given country [19], and some studies have established differences between rural and urban areas [20,21], while others have not been able to explore it [16].

There is substantially less information on the patterns of antibiotic resistance in patients in the community than in hospitalized patients, on whom clinical practice recommendations are based [7,11], and the behavior of prescription habits among large cities compared to those with fewer inhabitants is unknown. Therefore, we sought to compare the patterns of antibiotics prescriptions for outpatients between cities and municipalities in Colombia.

## 2. Results

A total of 182,397 people distributed throughout 187 different capital cities or municipalities who were prescribed an antibiotic were identified. Of these, 60.9% (*n* = 110,998) were women. The mean age was 37.1 ± 23.0 years (range: 0.08–108.75 years), with the following disbursement across age groups: <18 years (*n* = 42,129; 23.1%), 18–49 years (*n* = 83,726; 45.9%), 50–64 years (*n* = 31,293; 17.1%), 65–79 years (*n* = 18,918; 10.4%) and ≥ 80 years (*n* = 6331; 3.5%). A total of 60.5% (*n* = 110,327) resided in capital cities, and most of them were in the Caribbean Region (*n* = 91,290; 50.1%), followed by the Bogotá–Cundinamarca Region (*n* = 45,476; 24.9%), Central Region (*n* = 20,752; 11.4%), Pacific Region (*n* = 18,928; 10.3%), Eastern Region (*n* = 5079; 2.8%), and Amazon–Orinoco Region (*n* = 872; 0.5%).

A total of 37 different antibiotics were prescribed; 90.2% (*n* = 164,447) of the patients received a single antimicrobial, while 8.8% (*n* = 16,049) received two, and 1.0% (*n* = 1901) received three or more. A total of 93.5% (*n* = 170,619) of the patients were given antibiotics to be used orally, with tablets or capsules being the most frequently prescribed pharmaceutical dosage form (*n* = 149,660; 82.1%) (Table 1). The most commonly used groups of antibiotics were penicillins, followed by cephalosporins and fluoroquinolones (Table 1). The most frequently prescribed antibiotic was amoxicillin, followed by cephalexin and nitrofurantoin (Table 2). In terms of the AWaRe classification, 86.4% of the patients (*n* = 157,558) were prescribed Access antibiotics, 17.0% (*n* = 30,936) received Watch antibiotics, and 0.1% (*n* = 188) were prescribed Reserve antibiotics.

A total of 28.7% (*n* = 52,324) of all patients had some chronic pathology (Table 1). Of these, 89.3% (*n* = 46,754) had one to two pathologies, 9.5% (*n* = 4964) had three to four pathologies, and 1.2% (*n* = 606) had five or more pathologies. These comorbidities predominated in individuals aged 65 years or older (*n* = 17,093/25,107; 68.1%). The 10 most common comorbidities were arterial hypertension (13.5%; *n* = 24,669), diabetes (8421; 4.6%), hypothyroidism (4469; 2.5%), chronic gastritis (3174; 1.7%), dyslipidemia (2707, 1.5%), acne (2691; 1.5%), benign prostatic hyperplasia (2246; 1.2%), cancer (2026; 1.1%), irritable colon (1903; 1.0%), and osteoarthritis (1832; 1.0%).

### 2.1. Comparison between Capital Cities and Municipalities

Differences were found between capital cities and municipalities. In cities, it was more common to find prescriptions of a single antibiotic, tablet or capsule pharmaceutical dosage forms, penicillins, urinary antiseptics, tetracyclines, rifamycins and lincosamides, while in municipalities, it was more common to prescribe two or more antibiotics, injectable pharmaceutical dosage forms, cephalosporins, fluoroquinolones, macrolides, and aminoglycosides (Table 1). Amoxicillin was the most commonly used antibiotic in cities, while ciprofloxacin was the most commonly used antibiotic in municipalities, and there was no significant difference regarding cephalexin. The Access and Reserve groups of antibiotics were prescribed more frequently in cities, while the Watch group predominated in municipalities (Table 2).

### 2.2. Comparison among Age Groups

In each of the age groups, women and residents of cities represented the majority. For those under 18 years of age, the prescription of two or more antibiotics was less likely than for the other age groups. The pharmaceutical dosage forms of tablets or capsules predominated among those aged 18 and older, while powder to reconstitute to an oral solution was more frequently prescribed for those under 18 years of age. Penicillins and cephalosporins were prescribed more frequently to children under 18 years of age, while the prescription of fluoroquinolones and urinary antiseptics increased with advancing age and was most common for those older than 65 years. The prescription of tetracyclines showed a peak among patients between the ages 18 and 49 years, while prescriptions for macrolides, sulfonamides and aminoglycosides did not vary greatly among the different age groups. The prescription of antibiotics classified as Access decreased with patient age (*p* < 0.001), while the prescription of Watch and Reserve antibiotics increased with patient age (*p* < 0.001) (Table 3).

### 2.3. Comparison among Geographic Regions

The prescription of two or more antibiotics was found most frequently in the Caribbean Region. Penicillins, followed by cephalosporins, were the most commonly prescribed antibiotic groups in all regions of the country. In the Caribbean Region, the prescription of fluoroquinolones, macrolides and aminoglycosides was more common than in the other geographic regions, while tetracyclines were most often prescribed in Bogotá–Cundinamarca, and urinary antiseptics were most often prescribed in the Central and Pacific regions. The prescription of Access antibiotics predominated in the Bogotá–Cundinamarca Region, while the prescription of Watch antibiotics predominated in the Pacific and Caribbean regions (Table 4).

### 2.4. Multivariate Analysis

A binary logistic regression model was conducted using the prescription of antibiotics classified as Watch or Reserve (yes/no) as a dependent variable. The multivariate analysis found that prescriptions written in municipalities; the presence of one or more comorbidities; and the presence of urinary, respiratory or gastrointestinal pathologies increased the probability of receiving antibiotics classified as Watch and/or Reserve, while residing in the Bogotá–Cundinamarca, Central or Eastern regions and having dermatological, rheumatological, cardiovascular or neurological pathologies reduced this risk. Interaction terms were analyzed. There was an interaction between age and male sex; with higher probability of receiving Watch/Reserve antibiotics with increasing age (Table 5). Other interaction terms were tested but none was statistically significative (positive nor negative effects).

## 3. Discussion

The patterns of antibiotic prescription for outpatients of any age and sex were evaluated for nearly one-sixth of the population of Colombia to characterize the differences and similarities in the dispensing of these drugs according to geographic region, capital city or municipality and age group and the variables associated with the use of antibiotics classified as Watch or Reserve. In general, antibiotics were more often prescribed for women, a finding that has also been reported in the United States [1,5], Europe [17,22,23] and Africa [24]. The reason for this is that infectious diseases vary between men and women due to factors related to genetics, biology, and behavioral differences [25].

Penicillins were the most commonly used pharmacological group (38.6%), which is consistent with the findings of Goossens et al. in a multicenter study conducted in 28 countries in Europe and North America (38.9–45.7%) [26] and those of Elseviers et al. in 24 European countries (32.0–63.0%) [27]. Different studies have documented that in the United States and Malaysia, the use of penicillins predominated, but at a lower proportion (23.0% and 30.7%, respectively) [1,12]; this differs from the results of other American studies in which the most commonly prescribed antibiotics were fluoroquinolones (22.0–25.0%) [5,11], and with a study indicating that in Greece, the use of macrolides was more frequent (29.9%) [4].

Amoxicillin was the most commonly prescribed antibiotic (28.1%) in the present study, a finding consistent with results for countries in North America, Europe, Asia, and Africa (20.8–22.5%) [12,24,26] and with what was documented between 2006 and 2007 in Colombia (27.8%–29.7%) [28]. In the United States and Greece, the use of azithromycin (14.0–54.1%) [1,5,29] and clarithromycin (19.8%) [4], respectively, predominated. These variations in the prescription of antibiotics may be due to the existing epidemiological differences between countries in terms of the type and frequency of bacterial infections, the etiological agents involved, and their resistance patterns; variations may also be due to differences in clinical practice guidelines, which in some cases do not yield the same recommendations, as well as the marketing strategies of the pharmaceutical industry and the characteristics of each country’s health systems and accessibility to healthcare services.

Many studies have shown that the frequency of use and prescription patterns of antibiotics varies according to the geographical area of each country [1,5,17,22,23,30], a finding that was also evidenced in this analysis. No published studies comparing antibiotics use between the capital cities and municipalities of any country were found. However, in the Netherlands, de Jong et al. compared the use of antibiotics in patients who lived in rural areas with that of patients who lived in urban areas and found that the prevalence of antibiotic prescriptions was higher in rural areas (23.6% versus 20.2%, *p* < 0.001). In addition, the most commonly used antibiotic groups were penicillins and tetracyclines, which predominated in rural areas [20]. A study carried out in a region of Vietnam, for 3 consecutive days, evaluated the sales of medicines from 30 private pharmacies, 15 rural and 15 urban, finding that antibiotics represented 30% and 24% (*p* = 0.002) of dispensations from rural and urban pharmacies, with amoxicillin being the best-selling antibiotic (27% vs. 13%, *p* < 0.0001) [31]. In the United States, rural prescribers were more likely to formulate large amounts of antibiotics [32]. However, in a study conducted in Italy, urban municipalities were eight times more likely to have high prevalence rates of antibiotic use compared to rural municipalities (adjusted OR: 8.62; 95% CI: 4.06–18.30, *p* < 0.001) [21] and also in an investigation carried out in Sweden, the prescription of antibiotics was higher in urban areas [19]. In our study, significant differences were found in the prescription of some antibiotics, highlighting the use of fluoroquinolones, macrolides, aminoglycosides, and cephalosporins in the population residing in municipalities and penicillins and tetracyclines among those residing in cities. Among the factors that may influence these differences are the habits of prescribers and their degree of continuing education, as well as the prevalence of infectious pathologies, comorbidities, socioeconomic, cultural factors, and the educational level of patients, as well as the absence of local pharmacovigilance programs. Note that fluoroquinolones, macrolides and many cephalosporins belong to the Watch antimicrobial group.

In England, the majority of prescribed antibiotics were classified as Access (68.7%), followed by Watch (18.4%) and Reserve (0.4%) [33]; and in 70 high- and middle-income countries, the median rate of prescription of Access antibiotics was 76.3% (27.0–94.4%), compared with 12.3% (3.3–54.0%) for Watch antibiotics and 1.0% for Reserve antibiotics [34]. The Watch/Access ratio of these studies is consistent with that found in the present analysis of a sample of 20% of the prescriptions in Colombia: Access antibiotics were the most frequently prescribed group in the different geographical regions throughout the country, showing compliance with the WHO’s goal of using these antibiotics in a proportion greater than 60% in order to reduce microorganism resistance and achieve better therapeutic results [35].

Watch antibiotics have not been sufficiently characterized in epidemiological studies because this classification was first proposed by the WHO in 2017. In our analysis, this group was mainly represented by fluoroquinolones and macrolides, and the factors that were associated with increased use of this group of antibiotics were male sex, older age and number of comorbidities, probably because these drugs are indicated for prostatitis, sexually-transmitted infections, urinary tract infections, and upper and lower respiratory tract infections [36]. However, in this study, the specific clinical indications for the prescribed antibiotics could not be determined. Ciprofloxacin stood out as one of the most prescribed Watch antibiotics in this sample of Colombian prescriptions. In different studies, ciprofloxacin was used by 5.7–20.9% [1,4,5,12,24] of patients, and in Colombia between 2006–2007, it was used by 8.6% [28], slightly lower than its current rate of use in the country (9.2%). Its use, like that of Watch antibiotics in general, predominated in municipalities.

Some limitations in the interpretation of the results are recognized. There was no access to medical records to identify the clinical indications for the prescription of antibiotics, nor could it be established whether the antibiotics were taken as recommended, including whether they were taken for the minimum time. Likewise, on medical forms, the ICD-10 code corresponding to the infectious disease is not always recorded; thus, the quality of the prescriptions could not be determined, and information about antibiotics purchased outside the health system is unknown.

## 4. Materials and Methods

This was a cross-sectional study of the prescription of antibiotics for outpatient use. It compared antibiotics use between capital cities and municipalities in a group of Colombian patients from a population-based drug-dispensing database that collects information on approximately 8.5 million persons affiliated with the Colombian Health System and six insurers. The population covered by the database corresponds to 30.0% of the active population affiliated with the contributory regime and 6.0% with the subsidized regime, which corresponds to 16.3% of the Colombian population.

Patients of any age and sex who were treated at outpatient clinics and received antibiotics for outpatient use between January 1 and 31, 2020 were selected. A database was designed to collect the following groups of variables:Sociodemographic: Sex, age, insurance company, and place of dispensation (the term city was used to refer to all the capitals of Colombian departments (regions), and the term municipality was used for all other populations with fewer inhabitants).
Capital city and municipality of dispensation: See annex 1Geographic areas: The region of residence was categorized by department according to the regions of Colombia and considering the classification of the National Administrative Department of Statistics (Departamento Administrativo Nacional de Estadística—DANE) of Colombia as follows:Caribbean Region: Atlántico, Bolívar, Cesar, Córdoba, La Guajira, Magdalena, Sucre, San Andrés, Providencia, and Santa Catalina.Central Region: Antioquia, Caldas, Quindío, Risaralda, Caquetá, Huila, Tolima.Bogotá–Cundinamarca Region.Eastern Region: Boyacá, Meta, Norte de Santander, Santander, Arauca, Casanare.Pacific Region: Cauca, Chocó, Nariño, Valle del Cauca.Amazon–Orinoco Region: Amazonas, Guaviare, Guainía, Vaupés, Vichada, Putumayo.Chronic comorbidities: Identified from the main and secondary diagnoses reported using ICD-10 codes in the database between October 1, 2019, and January 31, 2020. Chronic comorbidities were grouped into four categories: no comorbidities and one, two and three or more pathologies. The following groups of diseases were considered:
Cardiovascular: High blood pressure, ischemic heart disease, tachyarrhythmias, heart failure, peripheral arterial disease.Endocrine: Diabetes, hypothyroidism, dyslipidemia, obesity, hyperthyroidism.Rheumatological: Osteoarthrosis, rheumatoid arthritis, osteoporosis, fibromyalgia, systemic lupus erythematosus, systemic sclerosis, ankylosing spondylitis.Renal: Chronic kidney disease.Psychiatric: Depression, anxiety, bipolar affective disorder, sleep disorders, psychosis.Neurological: Peripheral neuropathies, chronic pain, dementia, migraine, epilepsy, Parkinson’s disease, stroke, mental retardation.Digestive: Chronic gastritis, gastroesophageal reflux, constipation, cirrhosis, peptic ulcer, hepatitis, irritable colon.Respiratory: Chronic obstructive pulmonary disease, asthma.Urinary: Benign prostatic hyperplasia, urinary incontinence/overactive bladderSkin: Acne, psoriasis.Antibiotic groups according to the AWaRe classification (13) (see Table 6):Pharmaceutical dosage forms: Tablet, capsule, powder for reconstitution to oral solution, suspension, injectable solution, powder for inhalation.Number of antibiotics per patient: Grouped into three categories: One antibiotic, two antibiotics and three or more antibiotics received during the month of January.

The protocol was approved by the Bioethics Committee of Universidad Tecnológica de Pereira in the category of risk-free research (approval number: 05-090320). The ethical principles established by the Declaration of Helsinki were respected.

The data were analyzed with the statistical package SPSS Statistics, version 26.0 for Windows (IBM, USA). A descriptive analysis was performed with frequencies and proportions for the qualitative variables and measures of central tendency and dispersion for the quantitative variables. Quantitative variables were compared using Student’s t-test or ANOVA; categorical variables (i.e., age groups according to AWaRe classification) were compared using the X2 test. The Benjamini–Hochberg method was used to adjust for multiple hypotheses. An exploratory binary logistic regression model was fitted using the prescription of antibiotics classified as Watch or Reserve (yes/no). Sex, age groups and variables that were significantly associated in the bivariate analyses were used as covariates. Interaction terms and multicollinearity were also assessed (Variance inflation factor—VIF <10 as limit). The level of statistical significance adopted was *p* < 0.05.

## 5. Conclusions

The study results indicate that there are differences in the patterns of antibiotic prescription between capital cities and more rural municipalities; in particular, there is a greater proportion of Watch antibiotic use in municipalities than in cities, and there are differences in the prescription of penicillins, cephalosporins, fluoroquinolones, and macrolides. In general, the goals established by the WHO to reduce resistance to antimicrobials are being met, but it is necessary to implement continuing medical education measures to promote more homogeneous prescription patterns between cities and municipalities in different regions of the country. Studies of the use of antibiotics at the patient level in areas that prescribe a higher proportion of Watch antibiotics could provide data on the reasons for their use and whether it complies with treatment guidelines.

## Figures and Tables

**Table 1 antibiotics-09-00389-t001:** Comparison of sociodemographic, clinical and pharmacological characteristics of the patients included in the sample according to the place their live, Capital cities or small Municipalities (*p* values calculated by Chi-square/Fisher’s exact test, and adjusted using the Benjamini–Hochberg method)).

Variables	Total		Capital Cities		Municipalities		*p*
	*n* = 182,397	%	*n* = 110,327	%	*n* = 72,070	%	
Women	110,998	60.9	67,043	60.8	43,955	61,0	0.522
Men	71,399	39.1	43,284	39.2	28,115	39.0	0.522
Age (mean; SD)	37.06 ± 23.04		38.36 ± 22.35		35.06 ± 23.91		<0.001
Chronic comorbidities	52,324	28.7	37,866	34.3	14,458	20.1	<0.001
Cardiovascular	25,421	13.9	18,398	16.7	7,023	9.7	<0.001
Endocrine	16,050	8.8	11,805	10.7	4245	5.9	<0.001
Gastrointestinal	6910	3.8	5396	4.9	1514	2.1	<0.001
Neurological	5272	2.9	3995	3.6	1277	1.8	<0.001
Urinary	3907	2.1	2646	2.4	1261	1.7	<0.001
Psychiatric	3639	2.0	2693	2.4	946	1.3	<0.001
Rheumatological	3398	1.9	2488	2.3	910	1.3	<0.001
Respiratory	3106	1.7	2208	2.0	898	1.2	<0.001
Kidney	1495	0.8	889	0.8	606	0.8	0.522
Number of antibiotics per patient	-	-	-	-	-	-	-
1	164,447	90.2	100,639	91.2	63,808	88.5	<0.001
2	16,049	8.8	8772	8.0	7277	10.1	<0.001
3 or more	1901	1.0	916	0.8	985	1.4	<0.001
Pharmaceutical forms	-	-	-	-	-	-	-
Tablet or capsule	149,660	82.1	94,075	85.3	55,585	77.1	<0.001
Powder to reconstitute to oral solution	25,980	14.2	12,566	11.4	13,414	18.6	<0.001
Injectable	11,778	6.5	5886	5.3	5892	8.2	<0.001
Suspension	2328	1.3	989	0.9	1339	1.9	<0.001
Inhalation	14	0.0	12	0.0	2	0.0	0.428
Antibiotic groups	-	-	-	-	-	-	-
Penicillins	70,380	38.6	43,668	39.6	26,712	37.1	<0.001
With beta-lactamase inhibitors	2068	1.1	1486	1.3	582	0.8	<0.001
Cephalosporins	55,013	30.2	32,182	29.2	22,831	31.7	<0.001
Fluoroquinolones	19,813	10.9	10,190	9.2	9623	13.4	<0.001
Urinary antipseptics	14,839	8.1	9639	8.7	5200	7.2	<0.001
Tetracyclines	11,970	6.6	9085	8.2	2885	4.0	<0.001
Macrolides	9778	5.4	5269	4.8	4509	6.3	<0.001
Sulfonamides	8731	4.8	5338	4.8	3393	4.7	0.522
Aminoglycosides	6596	3.6	2436	2.2	4160	5.8	<0.001
Rifamycins	418	0.2	382	0.3	36	0.0	<0.001
Lincosamides	84	0.0	75	0.0	9	0.0	<0.001
Oxazolidinones	11	0.0	9	0.0	2	0.0	0.522
Phenicols	6	0.0	2	0.0	4	0.0	0.522
Glycopeptides	2	0.0	2	0.0	0	0.0	0.522

**Table 2 antibiotics-09-00389-t002:** Comparison of the antibiotics prescribed in the Capital cities and the small Municipalities, according to the AWaRe classification (*p* values calculated by Chi-square/Fisher’s exact test, and adjusted using the Benjamini–Hochberg method).

Antibiotics	Total		Capital Cities		Municipalities		*p*
	*n* = 182,397	%	*n* = 110,327	%	*n* = 72,070	%	
**Access**	157,558	86.4	96,942	87.9	60,616	84.1	<0.001
Amoxicillin	51,326	28.1	32,825	29.8	18,501	25.7	<0.001
Cephalexin	47,256	25.9	28,501	25.8	18,755	26.0	0.744
Nitrofurantoin	14,674	8.0	9493	8.6	5181	7.2	<0.001
Dicloxacillin	11,675	6.4	6526	5.9	5149	7.1	<0.001
Doxycycline	11,281	6.2	8526	7.7	2755	3.8	<0.001
Trimethoprim sulfamethoxazole	8731	4.8	5338	4.8	3393	4.7	0.744
Cephradine	6328	3.5	2558	2.3	3770	5.2	<0.001
Gentamicin	5803	3.2	2068	1.9	3735	5.2	<0.001
Benzathine penicillin G	3732	2.0	2365	2.1	1367	1.9	0.003
Ampicillin	2513	1.4	965	0.9	1548	2.1	<0.001
Amoxicillin + Clavulanate	1436	0.8	984	0.9	452	0.6	<0.001
Amikacin	787	0.4	360	0.3	427	0.6	<0.001
Tetracycline	689	0.4	559	0.5	130	0.2	<0.001
Ampicillin + sulbactam	631	0.3	502	0.5	129	0.2	<0.001
Clindamycin	84	0.0	75	0.1	9	0.0	<0.001
Moxifloxacin	69	0.0	63	0.1	6	0.0	<0.001
Phenoxymethyl penicillin	64	0.0	50	0.0	14	0.0	0.035
Tobramicin	14	0.0	12	0.0	2	0.0	0.471
Chloramphenicol	6	0.0	2	0.0	4	0.0	0.744
Limecyclin	6	0.0	6	0.0	0	0.0	0.526
Minocyclin	6	0.0	6	0.0	0	0.0	0.526
Amoxicillin + Sulbactam	5	0.0	4	0.0	1	0.0	0.744
Cefadroxil	2	0.0	1	0.0	1	0.0	0.999
**Watch**	30,936	17.0	16,503	15.0	14,433	20.0	<0.001
Ciprofloxacin	16,866	9.2	8636	7.8	8230	11.4	<0.001
Azithromycin	4015	2.2	1504	1.4	2511	3.5	<0.001
Clarithromycin	2852	1.6	1879	1.7	973	1.4	<0.001
Erythromycin	2707	1.5	1752	1.6	955	1.3	<0.001
Norfloxacin	2649	1.5	1301	1.2	1348	1.9	<0.001
Ceftriaxone	1560	0.9	1145	1.0	415	0.6	<0.001
Levofloxacin	289	0.2	208	0.2	81	0.1	<0.001
Spiramycin	232	0.1	147	0.1	85	0.1	0.999
Cefuroxime	103	0.1	91	0.1	12	0.0	<0.001
Cefpodoxime	1	0.0	1	0.0	0	0.0	0.999
Vancomycin	2	0.0	2	0.0	0	0.0	0.999
**Reserve**	188	0.1	167	0.2	21	0.0	<0.001
Fosfomycin	177	0.1	158	0.1	19	0.0	<0.001
Linezolid	11	0.0	9	0.0	2	0.0	0.876
**Others** (Rifaximin)	418	0.2	382	0.3	36	0.0	<0.001

**Table 3 antibiotics-09-00389-t003:** Distribution of the antibiotic prescription in the study sample according to the age group.

Variables	<18 Years		18–49 Years		50–64 Years		65–79 Years		≥80 Years	
	*n* = 42,129	%	*n* = 83,726	%	*n* = 31,293	%	*n* = 18,918	%	*n* = 6331	%
Women	21,125	50.1	55,217	65.9	19,863	63.5	10,970	58.0	3823	60.4
Men	21,004	49.9	28,509	34.1	11,430	36.5	7948	42.0	2508	39.6
Prescription in capital city	21,271	50.5	53,774	64.2	19,657	62.8	11,805	62.4	3820	60.3
Chronic comorbidities	3808	9.0	16,382	19.6	14,951	47.8	12,507	66.1	4676	73.9
Cardiovascular	799	1.9	5261	6.3	7984	25.5	8234	43.5	3143	49.6
Endocrines	474	1.1	4560	5.4	5533	17.7	4249	22.5	1234	19.5
Gastrointestinal	429	1.0	2907	3.5	1987	6.3	1197	6.3	390	6.2
Neurologic	335	0.8	2119	2.5	1050	3.4	987	5.2	781	12.3
Urinary	85	0.2	383	0.5	993	3.2	1500	7.9	946	14.9
Psychiatric	127	0.3	1284	1.5	1080	3.5	763	4.0	385	6.1
Rheumatological	29	0.1	527	0.6	1237	4.0	1196	6.3	409	6.5
Respiratory	692	1.6	454	0.5	514	1.6	895	4.7	551	8.7
Kidney	10	0	93	0.1	226	0.7	633	3.3	533	8.4
Number of antibiotics per patient	-	-	-	-	-	-	-	-	-	-
1	39,357	93.4	74,597	89.1	27,892	89.1	16,922	89.4	5679	89.7
2	2530	6.0	8158	9.7	3043	9.7	1747	9.2	571	9.0
3 or more	242	0.6	971	1.2	358	1.1	249	1.3	81	1.3
Pharmaceutical forms	-	-	-	-	-	-	-	-	-	-
Tablet or capsule	14,084	33.4	80,542	96.2	30,455	97.3	18,463	97.6	6116	96.6
Powder to reconstitute to oral solution	25,520	60.6	226	0.3	70	0.2	86	0.5	78	1.2
Injectable	1420	3.4	7075	8.5	1975	6.3	999	5.3	309	4.9
Suspension	2244	5.3	37	0.0	18	0.1	11	0.1	18	0.3
Inhalation	7	0.0	7	0.0	0	0.0	0	0.0	0	0.0
Antibiotic groups	-	-	-	-	-	-	-	-	-	-
Penicillins	20,142	47.8	31,122	37.2	11,556	36.9	5857	31.5	1603	25.3
With beta-lactamase inhibitors	494	1.2	777	0.9	379	1.2	278	1.5	140	2.2
Cephalosporins	15,169	36.0	24,403	29.1	8425	26.9	5201	27.5	1815	28.7
Fluoroquinolones	625	1.5	9277	11.1	4994	16.0	3672	19.4	1245	19.7
Urinary antiseptics	563	1.3	7804	9.3	3081	9.8	2286	12.1	1105	17.5
Tetracyclines	2109	5.0	7606	9.1	1378	4.4	694	3.7	183	2.9
Macrolides	2470	5.9	4111	4.9	1754	5.6	1072	5.7	371	5.9
Sulfonamides	2224	5.3	3681	4.4	1558	5.0	920	4.9	348	5.5
Aminoglycosides	678	1.6	3638	4.3	1341	4.3	724	3.8	215	3.4
Rifamycins	8	0.0	95	0.1	123	0.4	153	0.8	39	0.6
Lincosamides	2	0.0	48	0.1	21	0.1	8	0.0	5	0.1
Oxazolidinones	0	0.0	5	0.0	3	0.0	2	0.0	1	0.0
Phenicols	1	0.0	1	0.0	1	0.0	1	0.0	2	0.0
Glycopeptides	0	0.0	1	0.0	0	0.0	0	0.0	1	0.0
AWaRe classification	-	-	-	-	-	-	-	-	-	-
Access	39,382	93.5	72,752	86.9	25,717	82.2	14,795	78.2	4911	77.6
Watch	3255	7.7	14,350	17.1	6867	21.9	4817	25.5	1646	26.0
Reserve	1	0.0	55	0.1	40	0.1	58	0.3	34	0.5

**Table 4 antibiotics-09-00389-t004:** Comparison of different sociodemographic and pharmacological variables of patients who had one or more antibiotic prescriptions according to the Colombian geographic regions.

Variables	Caribbean Region		Bogota–Cundinamarca Region		Central Region		Pacific Region		Eastern Region		Amazonia–Orinoco Region	
	*n* = 91,290	%	*n* = 45,476	%	*n* = 20,752	%	*n* = 18,928	%	*n* =5079	%	*n* = 872	%
Women	55,737	61.1	28,028	61.6	12,458	60.0	11,249	59.4	3020	59.5	506	58.0
Men	35,553	38.9	17,448	38.4	8294	40.0	7679	40.6	2059	40.5	366	42.0
Prescription in capital city	40,170	44.0	39,439	86.7	15,554	75.0	10,545	55.7	3747	73.8	872	100.0
Chronic comorbidities	15,464	16.9	17,654	38.8	8752	42.2	8444	44.6	1926	37.9	84	9.6
Number of antibiotics per patient	-	-	-	-	-	-	-	-	-	-	-	-
1	80,727	88.4	42,254	92.9	18,932	91.2	17,037	90.0	4704	92.6	793	90.9
2	9306	10.2	2954	6.5	1643	7.9	1737	9.2	341	6.7	68	7.8
3 or more	1257	1.4	268	0.6	177	0.9	154	0.8	34	0.7	11	1.3
Pharmaceutical forms	-	-	-	-	-	-	-	-	-	-	-	-
Tablet or capsule	68,773	75.3	40,721	89.5	18,142	87.4	17,040	90.0	4328	85.2	656	75.2
Powder to reconstitute to oral solution	18,699	20.5	3387	7.4	1871	9.0	1228	6.5	625	12.3	170	19.5
Injectable	7343	8.0	1850	4.1	1057	5.1	1274	6.7	186	3.7	68	7.8
Suspension	1805	2.0	203	0.4	154	0.7	109	0.6	46	0.9	11	1.3
Inhalation	2	0.0	6	0.0	3	0.0	3	0.0	0	0.0	0	0.0
Antibiotic groups	-	-	-	-	-	-	-	-	-	-	-	-
Penicillins	31,685	34.7	20,166	44.3	8697	41.9	7323	38.7	2174	42.8	335	38.4
With beta-lactamase inhibitors	1098	1.2	410	0.9	216	1.0	288	1.5	50	0.0	6	0.7
Cephalosporins	31,158	34.1	11,536	25.4	5443	26.2	5210	27.5	1342	26.4	324	37.2
Fluoroquinolones	11,896	13.0	2980	6.6	1659	8.0	2740	14.5	460	9.1	78	8.9
Urinary antipseptics	6450	7.1	3960	8.7	2108	10.2	1828	9.7	437	8.6	56	6.4
Tetracyclines	3816	4.2	5082	11.2	1752	8.4	924	4.9	361	7.1	35	4.0
Macrolides	5624	6.2	1796	3.9	1107	5.3	976	5.2	232	4.6	43	4.9
Sulfonamides	4589	5.0	1843	4.1	1092	5.3	900	4.8	272	5.4	35	4.0
Aminoglycosides	5304	5.8	339	0.7	320	1.5	497	2.6	95	1.9	41	4.7
Rifamycins	51	0.1	197	0.4	90	0.4	71	0.4	8	0.2	1	0.1
Lincosamides	17	0.0	42	0.1	12	0.1	11	0.1	2	0.0	0	0.0
Oxazolidinones	1	0.0	1	0.0	5	0.0	4	0.0	0	0.0	0	0.0
Phenicols	2	0.0	0	0.0	1	0.0	3	0.0	0	0.0	0	0.0
Glycopeptides	0	0.0	2	0.0	0	0.0	0	0.0	0	0.0	0	0.0
AWaRe classification	-	-	-	-	-	-	-	-	-	-	-	-
Access	77,022	84.4	41,191	90.6	18,434	88.8	15,658	82.7	4489	88.4	764	87.6
Watch	17,981	19.7	5192	11.4	2946	14.2	3960	20.9	718	14.1	139	15.9
Reserve	24	0	73	0.2	31	0.1	53	0.3	6	0.1	1	0.1

**Table 5 antibiotics-09-00389-t005:** Multivariate analysis of the variables associated with the prescription of Watch and Reserve antibiotics in the study sample from Colombia, January, 2020.

Variables	*p* Value	OR	95% IC	
			Lower	Upper
Men	0.001	0.888	0.827	0.954
Age groups				
Age <18 years	<0.001	Reference	Reference	Reference
Age 18–49 years	<0.001	2.656	2.515	2.805
Age 50–64 years	<0.001	3.464	3.256	3.684
Age ≥65 years	<0.001	3.952	3.699	4.221
Age groups by Men				
Age 18–49 years	<0.001	1.174	1.082	1.274
Age 50–64 years	0.001	1.161	1.059	1.272
Age ≥65 years	<0.001	1.234	1.124	1.354
Prescription in municipalities	<0.001	1.269	1.235	1.304
Caribbean region	0.230	1.120	0.930	1.349
Central region	<0.001	0.663	0.549	0.801
Bogota-Cundinamarca region	<0.001	0.547	0.453	0.659
Eastern region	0.001	0.700	0.573	0.857
Pacific region	0.365	0.916	0.759	1.106
No chronic comorbidities	<0.001	Reference	Reference	Reference
One chronic comorbidity	<0.001	1.142	1.078	1.210
Two or more chronic comorbidities	<0.001	1.282	1.151	1.429
Cardiovascular pathologies	<0.001	0.874	0.822	0.929
Endocrine pathologies	0.199	0.962	0.903	1.025
Gastrointestinal pathologies	<0.001	1.320	1.227	1.420
Neurological pathologies	<0.001	0.844	0.777	0.916
Dermatological pathologies	<0.001	0.243	0.197	0.300
Psychiatric pathologies	0.156	0.938	0.855	1.028
Respiratory pathologies	<0.001	1.446	1.317	1.588
Rheumatological pathologies	<0.001	0.824	0.749	0.907
Urinary pathologies	<0.001	1. 586	1.460	1.722

**Table 6 antibiotics-09-00389-t006:** Antibiotics available in the Colombian Health System categorized according to the AWaRe classification.

AWaRe Classification	ATC Subgroup	Antimicrobials	
**Access**	J01A	Tetracyclines	Tetracyclin, doxycyclin, minocyclin, lymecyclin
	J01B	Amphenicols	Chloramphenicol
	J01C	Beta-lactam antibacterials—penicillins	Phenoximethyl penicillin V, penicillin G benzatin, amoxicillin, amoxicillin + clavulanic, ampicillin, ampicillin + sulbactam, dicloxacillin
	J01D	Other beta-lactam antibacterials	Cefalexin, cefradin, cefadroxil
	J01E	Sulfonamides and trimethoprim	Trimethoprim + sulfamethoxazol
	J01F	Macrolides, lincosamides and streptogramins	Clindamycin
	J01G	Aminoglycoside antibacterials	Gentamicin, amikacin, tobramycin
	J01X	Other antibacterials	Nitrofurantoin
**Watch**	J01D	Other beta-lactam antibacterials	Cefuroxime, ceftriaxone, cefpodoxime
	J01F	Macrolides, lincosamides and streptogramins	Erythromycin, clarithromycin, azithromycin, spiramycin
	J01M	Quinolone antibacterials	Ciprofloxacin, norfloxacin, levofloxacin, moxifloxacin
	J01X	Other antibacterials	Vancomycin
**Reserve**	J01X	Other antibacterials	Linezolid
			Fosfomycin
**Other**	A07A	Intestinal antiinfectives	Rifaximin

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
