# Peer review of "Patterns of Antibiotic Prescription in Colombia: Are There Differences between Capital Cities and Municipalities?"

_antibiotics, 2020, doi:10.3390/antibiotics9070389_

Round 1

Reviewer 1 Report

The study is interesting but needs some important improvements.

Can the Authors explain the acronym AWaRe and WHO also in the Abstract?

Introduction

The Introduction is not complete. This is a study concerning mainly the use of antibiotics of people living in different areas and to better introduce the theme, I expected some more related material and references. Moreover, I advise the Authors to be more exhaustive adding other information about resistance in the most common antibiotics that are missing.

62-62. Are the medications defined by the WHO? References are needed. Can the Authors illustrate more details about the concept of Access, Watch and Reserve?

M&M

The formal authorisation of the Bioethics Committee needs an approval number. Can the Authors add it?

Please, remove “no personal patient data were included”. The pathologies included are private patient information.

244: Is not clear the meaning. Which municipalities were considered?

Is not written how many people are involved in this study (182,397?)

Results

Table 1: The sum of women in capital cities (67053) and municipalities (43955) is 111.008.

Table 1,3,4: Why is not shown also the men information?

Discussion

161: This sentence is too long. Please, reword it. I do not understand how 182,397 people can be nearly one-fifth of the entire population.

The Introduction should introduce the general information of the topic considered, while the Discussion should be an “amalgam” which introduce/explain your results taking in consideration data from other studies and see if there is a relationship.  

In this Discussion I did not appreciate it, and I often see a list of other studies that could stay in the Introduction.

I would invite the Authors to renovate and get better the Introduction and Discussion.

Author Response

20-Jun-2020

Dear Editors

Antibiotics

Manuscript ID Reference 828558: Patterns of antibiotic prescription in Colombia: Are there differences between capital cities and municipalities?

Answer for Reviewers' Comments to Author:

Reviewer comments for: Reference 828558

Referee 1

Comment

Author Response

Text Insertion (if applicable)/ page/ line number of change

General Comments:

n/a

Reviewer 1

The study is interesting but needs some important improvements.

Can the Authors explain the acronym AWaRe and WHO also in the Abstract?

is added in the abstract

Page 1. Lines 20-22

The Introduction is not complete. This is a study concerning mainly the use of antibiotics of people living in different areas and to better introduce the theme, I expected some more related material and references. Moreover, I advise the Authors to be more exhaustive adding other information about resistance in the most common antibiotics that are missing.

New information about the main microorganisms is added with which there are resistance problems and to which antibiotics

Page 2.

62-62. Are the medications defined by the WHO? References are needed. Can the Authors illustrate more details about the concept of Access, Watch and Reserve?

Expanded with new information in the introduction

Page 1, Lines: 69-73

M&M

The formal authorisation of the Bioethics Committee needs an approval number. Can the Authors add it?

Approval number is added

Page 12, line 324

Please, remove “no personal patient data were included”. The pathologies included are private patient information.

Withdraw no personal patient data were included

Page 12, line 325

244: Is not clear the meaning. Which municipalities were considered?

A supplementary annex is added with the list of capital cities and municipalities included

Supplementary Annex 1.

Is not written how many people are involved in this study (182,397?)

In fact, there were 182,397 patients distributed in 187 cities and municipalities.

Page 2, line: 99

Results

Table 1: The sum of women in capital cities (67053) and municipalities (43955) is 111.008.

Is corrected

Page 3, table 1

Table 1,3,4: Why is not shown also the men information?

Men's data is aggregated.

Tables 1, 3, and 4.

Discussion

161: This sentence is too long. Please, reword it. I do not understand how 182,397 people can be nearly one-fifth of the entire population.

As you can see in methods, the analyzed population corresponds to 16.3% of Colombians, so the analysis of antibiotics in this study is approximately one fifth of the country's population.

Page 3.

The Introduction should introduce the general information of the topic considered, while the Discussion should be an “amalgam” which introduce/explain your results taking in consideration data from other studies and see if there is a relationship.  In this Discussion I did not appreciate it, and I often see a list of other studies that could stay in the Introduction. I would invite the Authors to renovate and get better the Introduction and Discussion.

Introduction and discussion are improved.

Page 1.

Reviewing: 2

Considering your paper reported about patterns of antibiotic prescription in Colombia, my major comment on this study is that the manuscript does not provide a convincing case regarding the really new contributions of pharmacological area.

Thanks for checking it out. We made corrections to improve it as suggested by the other reviewers.

Reviewing: 3

The authors investigate the patterns of antibiotics prescriptions for outpatients between cities and municipalities in Colombia. They made several separate elementary statistical analysis from comparison among demographic factors.

At the end, they concluded that there is a greater proportion of Watch antibiotic use in municipalities than in cities.

Overall, their statistical analysis needs improvement. Many analysis are carried out in a shallow level, not relevant to their main claim. The authors needs significant efforts to provide coherent analysis. For example, The prescribed antibiotics are compared without consideration of underlying pathologies. Perhaps, the claimed differences results from differences in age groups and/or pathologies between municipalities and cities.

Unfortunately, the diagnosis for which the antibiotics were prescribed could not be analyzed, as stated in the limitations in the discussion section. The other variables were explored in the multivariate analysis. In this regard, we made the corrections made by the reviewer in the following commentary

For decent analysis, the author should consider interaction terms in the logistic model and/or thre-way tables for comparison of demographic variables.

We considered the interaction terms in the logistic model. We tested interactions between variables such as age categories and sex. Other interaction terms among the variables included in the multivariate model were also explored but were not finally included due to collinearity. This is now briefly described both in results and methods

Page 8, lines 174-184. Table 5

Also, many statistical procedures are employed for testing multiple hypotheses simultaneously (e.g. Tables 1, 2, etc). However, the authors didn’t adjust the procedures for multiple hypotheses testing framework. They should use the family-wise significance level or false discovery rate for their testing

Our aim was to describe the entire population obtained in the database, so we did not used any sampling / inference process. Thus, in order to avoid testing for multiple hypotheses, we eliminated the p values comparing capital cities vs municipalities. No further analyses were made in this regard.

Pages 4, 5. Table 2

Minor Comments

• What is the rationale for the imbalanced age groups? Can you use it as a continuous variable?

The main continuous variable in the multivariate model was age. We used age groups to better show the difference between them, rather than analyzing the OR change/ unit (year). The selected age groups are those commonly used in descriptive studies, including patients under the age of 18 (children, adolescents) and on the other hand those >65 years (elderly population).

Table 3

How percentages in tables are calculated? (e.g. marginal, conditional, joint, etc.).

The percentages in tables are conditional (% in columns, within each group).

• L132 P5: ”The prescription of antibiotics classified as Access decreased with patient age, 133 while the prescription of Watch and Reserve antibiotics increased with patient age (Table 3)”. Is this statistically significant?

Yes, the change was significative. The p value in this regard was included in the section 2.2.

Page 5, line 134

Page 6, lines 156-157

Section 2.4: What is the multivariate analysis?

We added a sentence in this section in order to state the multivariate analysis used. Also, the description of the multivariate analysis in the methods section was updated.

Page 6, lines 174-175

The authors

Reviewer 2 Report

Considering your paper reported about patterns of antibiotic prescription in Colombia, my major comment on this study is that the manuscript does not provide a convincing case regarding the really new contributions of pharmacological area.

Author Response

(The authors gave the same response as above.)

Author Response

(The authors gave the same response as above.)

Reviewer 3 Report

The authors investigate the patterns of antibiotics prescriptions for outpatients between cities and municipalities in Colombia. They made several separate elementary statistical analysis from comparison among demographic factors.
At the end, they concluded that there is a greater proportion of Watch antibiotic use in municipalities than in cities.
Overall, their statistical analysis needs improvement. Many analysis are carried out in a shallow level, not relevant to their main claim. The authors needs significant efforts to provide coherent analysis. For example, The prescribed antibiotics are compared without consideration of underlying pathologies. Perhaps, the claimed differences results from
differences in age groups and/or pathologies between municipalities and cities. For decent analysis, the author should consider interaction terms in the logistic model and/or thre-way tables for comparison of demographic variables.
Also, many statistical procedures are employed for testing multiple hypotheses simultaneously (e.g. Tables 1, 2, etc).
However, the authors didn’t adjust the procedures for multiple hypotheses testing framework. They should use the
family-wise significance level or false discovery rate for their testing.
Minor Comments
• What is the rationale for the imbalanced age groups? Can you use it as a continuous variable?
• How percentages in tables are calculated? (e.g. marginal, conditional, joint, etc.).
• L132 P5: ”The prescription of antibiotics classified as Access decreased with patient age, 133 while the prescription
of Watch and Reserve antibiotics increased with patient age (Table 3)”. Is this statistically significant?
• Section 2.4: What is the multivariate analysis?

Author Response

(The authors gave the same response as above.)

Author Response

(The authors gave the same response as above.)

Round 2

Reviewer 1 Report

In the reviewed version, the Authors have adopted my suggestions.

I have just a consideration to do. How can 182,397 people be nearly one-fifth of the entire Colombian population that is of about 50 million?

Author Response

*In the reviewed version, the Authors have adopted my suggestions.

I have just a consideration to do. How can 182,397 people be nearly one-fifth of the entire Colombian population that is of about 50 million?

R/ The Colombian population is around 49 millions.  The population-based drug-dispensing database that collects information on approximately 8.5 million persons which corresponds to 16.3% of the Colombian population or one-sixth of the people.  When we identify 182,397 users of antibiotics, it means that it reflects the consumption of antibiotics in that period of a sixth of the population We corrected (line 192).

The authors

Reviewer 2 Report

Your revision was adequately addressed.

Author Response

*Your revision was adequately addressed.

R/ Thank you

Reviewer 3 Report

Dropping p-values from the manuscript may not a proper way to handle a non-random sample. With no statistical inference, contingency tables provide no supporting evidence to their claim. 

The logistic regression is based on the likelihood of stochastic binary outcome. If the authors believe no statistical inference should be performed, p-values and confidence intervals are invalid and the logistic regression itself is an sub-optimal solution. I believe there are a handful of modeling framework employed in machine learning context.

Lastly, they chose not to include interaction terms due to multicollinearity. This questioned me about their model selection procedure. Rather than cherry-picking a few predictors, which will lead to a biased results, they should use a proper model selection procedure that suits well with their classification model.

Author Response

Reviewer 3

*Dropping p-values from the manuscript may not a proper way to handle a non-random sample. With no statistical inference, contingency tables provide no supporting evidence to their claim.

R/ We believe there has been some misunderstanding regarding the reviewer’s observations and how we have interpreted them. In the previous comments the reviewer stated that we made unadjusted multiple hypotheses. We considered dropping p-values because the database analyzed comprised the entire population with antibiotics during the study period and we did not make any sample – our intention was to describe the different variables according to cities/municipalities. This database is one of the best representations of the country population (with information regarding more than 8.5 million patients, as stated in the materials and methods section), and it has been used in many other pharmacoepidemiological papers.

However, with this new comment, we have decided to conduct the suggested observations and have thus re-analyzed the information on tables 1 (lines 118-122) and 2 (lines 123-126) in order to adjust the p values using the Benjamini-Hochberg method. This will fulfill previous request and support the evidence provided in this contingency tables.

This has been also added in the methods section (lines 324. 325).

*The logistic regression is based on the likelihood of stochastic binary outcome. If the authors believe no statistical inference should be performed, p-values and confidence intervals are invalid and the logistic regression itself is an sub-optimal solution. I believe there are a handful of modeling framework employed in machine learning context.

R/ We have decided to adjust the analyzes and have added the new adjusted p-values for the initial tables, as expressed in the previous response. Adding other multivariate models seems to deviate the initial objective of the study and the proposed methods.

*Lastly, they chose not to include interaction terms due to multicollinearity. This questioned me about their model selection procedure. Rather than cherry-picking a few predictors, which will lead to a biased results, they should use a proper model selection procedure that suits well with their classification model.

R/ We in fact included interaction terms. Only one (age by sex) was significative, the other interactions were run but none was important or presented a modifying effect in the model (positive nor negative). The variables included in the original model (without interactions) do not have multicollinearity. We have changed the sentence regarding this in the section 2.4. (line 181-184).

The authors

Round 3

Reviewer 3 Report

Thanks to the authors for their response which have addressed most of my
concerns. The manuscript has been improved. I recommend to accept the manuscript for publication.